# Role of p*K*_A_ in Charge Regulation and Conformation of Various Peptide Sequences

**DOI:** 10.3390/polym13020214

**Published:** 2021-01-09

**Authors:** Raju Lunkad, Anastasiia Murmiliuk, Zdeněk Tošner, Miroslav Štěpánek, Peter Košovan

**Affiliations:** Department of Physical and Macromolecular Chemistry, Faculty of Science, Charles University, 128 43 Prague, Czech Republic; raju.lunkad@natur.cuni.cz (R.L.); anastasiia.murmiliuk@natur.cuni.cz (A.M.); zdenek.tosner@natur.cuni.cz (Z.T.); miroslav.stepanek@natur.cuni.cz (M.Š.)

**Keywords:** peptide, ampholyte, ionization, acid-base equilibrium, charge regulation, simulation, polyelectrolyte, coarse-grained, constant-pH

## Abstract

Peptides containing amino acids with ionisable side chains represent a typical example of weak ampholytes, that is, molecules with multiple titratable acid and base groups, which generally exhibit charge regulating properties upon changes in pH. Charged groups on an ampholyte interact electrostatically with each other, and their interaction is coupled to conformation of the (macro)molecule, resulting in a complex feedback loop. Their charge-regulating properties are primarily determined by the $pK_{A}$ of individual ionisable side-chains, modulated by electrostatic interactions between the charged groups. The latter is determined by the amino acid sequence in the peptide chain. In our previous work we introduced a simple coarse-grained model of a flexible peptide. We validated it against experiments, demonstrating its ability to quantitatively predict charge on various peptides in a broad range of pH. In the current work, we investigated two types of peptide sequences: diblock and alternating, each of them consisting of an equal number of amino acids with acid and base side-chains. We showed that changing the sequence while keeping the same overall composition has a profound effect on the conformation, whereas it practically does not affect total charge on the peptide. Nevertheless, the sequence significantly affects the charge state of individual groups, showing that the zero net effect on the total charge is a consequence of unexpected cancellation of effects. Furthermore, we investigated how the difference between the $pK_{A}$ of acid and base side chains affects the charge and conformation of the peptide, showing that it is possible to tune the charge-regulating properties by following simple guiding principles based on the $pK_{A}$ and on the amino acid sequence. Our current results provide a theoretical basis for understanding of the complex coupling between the ionisation and conformation in flexible polyampholytes, including synthetic polymers, biomimetic materials and biological molecules, such as intrinsically disordered proteins, whose function can be regulated by changes in the pH.

## 1. Introduction

Charge regulation in peptides, and more generally in weak ampholytes, is important not only in biological systems but also in the applications of bio-inspired or synthetic pH-responsive materials. Changes in pH can be used to control enzyme activity or protein aggregation [1,2], to control the release of anti-cancer drugs [3], or protein sequestration in polyelectrolyte complexes, brushes or hydrogels [4,5,6,7]. If such a system contains both weak acid and weak base groups, then their ionisation states may change simultaneously upon a change in pH. The generic picture is provided by the Henderson-Hasselbalch equation, relating the degree of ionisation, α, of a weak acid or base group with its $pK_{A}$ and pH
(1)$$\begin{array}{c} \mathrm{pH}-pK_{A}^{\mathrm{acid}}=\log_{10}\frac{\alpha }{1-\alpha }\\ \mathrm{pH}-pK_{A}^{\mathrm{base}}=\log_{10}\frac{1-\alpha }{\alpha }, \end{array}$$
where $pK_{A}^{\mathrm{acid}}$ and $pK_{A}^{\mathrm{base}}$ are the acidity constants of the acid and base groups, corresponding to the generic acid-base reactions of an acid HA and base B [8]
(2)$$HA\;⇌\;A^{-}\;+\;H^{+}$$
(3)$${\mathrm{BH}}^{+}\;⇌\;B\;+\;H^{+}.$$

Based on the Henderson-Hasselbalch equation, one can determine the total charge on an ampholyte as
(4)$$\begin{array}{c} z\left(\mathrm{pH}\right)=\sum_{i}\alpha_{i}\left(\mathrm{pH}\right)z_{i}, \end{array}$$
where $z_{i}$ is the valency of an individual ionisable group *i*. Models based on such an ideal approach allow to estimate the isoelectric point of proteins and peptides based on their amino acid composition [9]. Experimental observations correlate with these predictions, however, they also exhibit clear systematic deviations, apparently caused by interactions between acid and base groups [10]. In the case of rigid objects with many titratable groups, such as globular proteins, colloids or solid surfaces, these deviations can be attributed to the local electrostatic potential, which depends on the location and ionisation states of all ionisable groups [11,12,13]. As a rule of thumb, if an acid or base group is bound to like-charged groups, then it is ionised less than in the ideal case. If it is bound to oppositely charged groups, then it is ionised more than in the ideal case. Models based on solving the Poisson-Boltzmann equation have been successful in predicting these deviations in globular proteins or in colloids [12,14,15,16,17,18]. Consequently, charge regulation in such systems can be considered well understood.

In many relevant systems, charge regulation is accompanied by conformational changes. In such case, not only the ionisation states of acid and base groups but also their location in space change upon a change in pH. Examples of such systems include polyampholyte gels which change their swelling upon a change in pH [19,20], or polyzwitterions which change their interactions with charged colloids upon a change in the pH [21,22,23,24]. Peptide-based pH-responsive systems and certain therapeutic peptides exhibit similar features [25,26,27]. Lastly, the term nanobuffering has been coined recently to describe systems in which the charge regulation is coupled to a conformational change in such a way, that local environment in the vicinity of the nanobuffer remains approximately constant even though pH of the solution is changing [28,29]. Systematic understanding of flexible ampholytes can no longer be achieved by solving the Poisson-Boltzmann equation for a rigid object. However, it can be achieved by using molecular simulations and suitable coarse-grained models. Furthermore, it requires the generic principles of charge regulation in rigid ampholytes to be combined with the understanding of coupling between ionisation and conformation of long flexible polyelectrolytes. Even though partial answers to each of the above problems have been available in the literature, a holistic picture of the ionisation of flexible ampholytes seems to be lacking.

Molecular simulations using coarse-grained models have been previously applied to study charge regulation in synthetic weak polyelectrolyte homopolymers, consisting of weak acid or weak base groups [8]. From these simulations we have learned that ionisation of weak polyelectrolytes increases upon an increase in pH but it is suppressed as compared to the ideal case, as a consequence of electrostatic repulsion between like-charged groups [8,30,31,32,33,34,35,36,37,38,39,40,41]. This increase of ionisation is accompanied by expansion of the polyelectrolyte chain, resulting in a dramatic increase of the end-to-end distance or gyration radius. Both these effects gain on significance as the chain length of the polyelectrolyte is increased, however, they saturate at a certain chain length, N≳50. These results are supported by various approximate theories and by experimental observations [42,43]. Even though this general understanding is well established, its extrapolation to weak ampholytes is not straightforward.

Compared to simulations and theories for weak polyelectrolyte homopolymers, analogous results for weak polyampholytes are much more scarce [8]. Several studies used generic bead-spring models to investigate the role of $pK_{A}$ of acid and base groups in the ionisation and conformational response of flexible ampholytes in dilute solutions [44,45,46,47]. From these studies we learned that the presence of oppositely charged groups in the ampholyte tends to enhance their ionisation as compared to weak polyelectrolyte consisting only of like-charged groups. This general trend in ionisation response seems to be rather robust and surprisingly independent of the sequence: block-like or alternating. Additionally, these simulations have shown that electrostatic attraction between the oppositely charged groups results in a collapsed conformation of the ampholyte close to the isoelectric point, when both acid and base groups are ionised. Similar enhancement of ionisation has been observed in the case of polyelectrolytes interacting with oppositely charged nanoparticles [48,49,50]. In contrast with that, far from the isoelectric point the simulations revealed tadpole conformations of diblock sequences, and more uniformly stretched conformations of multiblock or alternating sequences. However, in each of the above studies, the authors have chosen one particular set of values of $pK_{A}^{\mathrm{acid}}$ and $pK_{A}^{\mathrm{base}}$. Thus, it remained unclear whether the overall enhancement of ionisation and conformational collapse was a generic feature of weak ampholytes, or whether it was a specific effect for the given combination of parameters.

Additional knowledge about charge regulation in weak ampholytes can be inferred indirectly from studies addressing related topics, such as structure and aggregation of ampholytes, peptides or disordered proteins without charge regulation. For example, the simulations and experiments by Sing, Perry and coworkers used peptides as model ampholytes with a well-defined sequence to demonstrate how the position of charged groups in the sequence affects their tendency to aggregate and form coacervates, suggesting that block-like structure and compatibility of sequences both promote aggregation [51]. The role of charge regulation in the association behaviour of oppositely charged polyelectrolytes was investigated by Rathee et.al. [52,53], showing that cooperative effect in the ionisation of both acid and base groups can be significant, especially if their $pK_{A}$ values are close to each other. Other simulation studies using generic coarse-grained models of ampholytes came to similar conclusions [46,54,55]. Simulation studies of real peptides or disordered proteins also revealed the importance of position of charged groups within the sequence [56,57,58]. However, in these studies the electrostatic interactions were just one of many factors contributing to the conformation and structure of the investigated peptide sequences.

Thus, even though there is an ample evidence from biomaterials and biochemical research that the sequence of charges plays an important role, a systematic investigation of its coupling to charge regulation and to the possible combinations of values of $pK_{A}^{\mathrm{acid}}$ and $pK_{A}^{\mathrm{base}}$ seems to be missing in the literature. In the current study we bridge this gap, combining the existing knowledge from polymer science with a simulation model of specific peptide sequences. The latter has been validated in our previous study where we used two model peptides, each of which consisted of a block of five acid and five base groups, with two different combinations of $pK_{A}$ values: $\Delta pK_{A}\equiv pK_{A}^{\mathrm{base}}-pK_{A}^{\mathrm{acid}}\gg 0$ and $\Delta pK_{A}\approx 0$ [59]. We observed not only that the enhanced ionisation of both acid and base groups seems to be a generic effect independent of $\Delta pK_{A}$ but also that computer simulations using simple coarse-grained models quantitatively agree with experiments, without any adjustable parameters. In the current study we build on the previous work, investigating the ionisation response and conformational characteristics of a broader set of peptides with various values of $\Delta pK_{A}$. In addition to the two peptide compositions investigated previously, we consider other peptides with $\Delta pK_{A}\ll 0$, $\Delta pK_{A}\approx 0$, and a hypothetical peptide with $\Delta pK_{A}=0$. Admittedly, a peptide with $\Delta pK_{A}=0$ could not be synthesized using natural amino acids. However, exploring the situation with $\Delta pK_{A}$ exactly equal to zero is important in order to generalize our conclusions to other ampholytes. For all these peptides we compare the simulation predictions for diblock and alternating sequences, showing that their ionisation response to pH is barely distinguishable because of cancellation of effects, while their conformational response differs qualitatively. The selected peptide sequences were rather short in order to prevent aggregation and to allow reliable experimental determination of their ionisation in a broad range of pH, especially around the isoelectric point. Therefore, in the final part of the current manuscript we investigate the role of chain length while keeping the sequence type unchanged (diblock or alternating).

## 2. Materials and Methods

### 2.1. Coarse-Grained Simulations

#### 2.1.1. Simulation Model

To represent the peptides, we used a coarse-grained bead-spring model derived from the one developed in our previous work [59]. The parameters of this model were adjusted to approximately reproduce all-atom simulations, as described in Section 2.1.3. The simulation box contained 10 peptide chains, where each peptide chain was composed of 5 acidic amino acids and 5 basic amino acids. Each amino acid was represented by a central bead C, and an acidic side chain A, or basic side chain B, as shown in Figure 1. The charge on the A and B beads was set to zero in the non-ionised state, and to ±1e in the ionised state, respectively. We chose five natural amino acids and two hypothetical amino acids with various values of $pK_{A}$, listed in Table 1. We combined these amino acids into various peptides listed in Table 2, which we distinguish based on the values of $\Delta pK_{A}\equiv pK_{A}^{\mathrm{base}}-pK_{A}^{\mathrm{acid}}$. For each combination of amino acids, we modeled a diblock and alternating sequence of acid and base groups. For completeness, we would like to mention that in the early stages of this work we considered a simpler model of the peptides, in which each amino acid was represented by just one bead. However, this model could not be tuned to reproduce the parameters obtained from the all-atom simulations, and its predictions of charge regulation did not match the experimental results as closely as the predictions from the two-bead model. A short discussion of these differences is provided in the Appendix A. Nevertheless, both types of models provide very similar predictions of ionisation response and they could be considered nearly equivalent, unless a precise quantitative comparison is sought. To keep our current results consistent with our earlier publication [59], we present here only the results obtained using the two-bead model depicted in Figure 1.

The simulation box length L=25.513nm corresponds to the concentrations of amino acid groups [Glu]=[His]=[Asp]=[Lys]=[Tyr]=[hypotheticalacid]=[hypotheticalbase]=5mM. By adding 100 additional salt ion pairs we kept the ionic strength at an approximately constant value of 10mM. Concentrations of the peptide and ionic strength were chosen based on our previous work, matching the conditions of potentiometric titrations and CZE experiments [59]. The ESPResSo simulation software was used for simulations of the CG model [61].

#### 2.1.2. Constant-pH Method

We used the constant-pH ensemble method to account for the weak acid and base reactions Equations (Equation 2) and (Equation 3). In both reactions, H^+^ was inserted or deleted at a random position in the system, and the new state was accepted with the probability,
(5)$$P_{\mathrm{acc}}^{\xi }=\min\left[1,\exp\left(-\beta \Delta U_{\mathrm{on}}+\xi \left(\mathrm{pH}-pK_{A}\right)\ln\left(10\right)\right)\right],$$
where $\beta =1/k_{B}T$, $\Delta U_{\mathrm{on}}=U_{n}-U_{o}$ is the energy difference between the new state (n) and the old state (o), and ξ=±1 for the forward or reverse direction of the reaction. The $pK_{A}$ and pH were input parameters, while the degree of ionisation was output of the simulation. The reactions were implemented as a Monte Carlo procedure, as described in Reference [8].

#### 2.1.3. Interaction Potentials

The connectivity of the bonds in the peptide model was represented by the harmonic potentials,
(6)$$U_{h}\left(r\right)=-\frac{k_{h}}{2}{\left(r-b\right)}^{2},$$
with the stiffness constant $k_{h}=400\;k_{B}T{\mathrm{nm}}^{-1}$ common to all bonds. The parameter *b* determines the equilibrium bond length. The bond length between central beads was $b_{\mathrm{CC}}=0.382\;\mathrm{nm}$ for all amino acid pairs. The bond lengths between central and side-chain beads, $b_{\mathrm{AC}}$ or $b_{\mathrm{BC}}$, for all amino acid pairs are listed in Table 1. These values were determined from all-atom (AA) simulations, as detailed in the Appendix A.

The short-range excluded volume interactions between all bead pairs were modeled using the Weeks-Chandler-Andersen (WCA) potential,
(7)$$U_{\mathrm{WCA}}\left(r\right)=\left\{\begin{array}{cc} 4\epsilon \left[{\left(\frac{\sigma }{r}\right)}^{12}-{\left(\frac{\sigma }{r}\right)}^{6}\right]+\epsilon & \;r\leq r_{\mathrm{cut}}\\ 0 & \;r>r_{\mathrm{cut}}, \end{array}\right.$$
where $\epsilon =1\;k_{B}T$, σ=0.355nm and $r_{\mathrm{cut}}=0.4\;\mathrm{nm}$, which defines the effective particle size.

The long-range electrostatic interaction between two charged beads, *i*, and *j* was represented by the Coulomb potential,
(8)$$U_{ij}\left(r\right)=z_{i}z_{j}k_{B}T\frac{l_{B}}{r}=\frac{1}{4\pi \epsilon_{0}\epsilon_{r}}\frac{z_{i}z_{j}e^{2}}{r},$$
where *z* is the valency, *e* is the elementary charge, $\epsilon_{0}$ is the permittivity of free space and $\epsilon_{r}$ is the relative permittivity. This interaction was handled via the P3M algorithm. The Bjerrum length, $l_{B}=0.71\;\mathrm{nm}$ was set to its approximate value in aqueous solutions at ambient temperature.

#### 2.1.4. Simulation Protocol and Data Analysis

All Simulations were performed using the Langevin dynamics. The Langevin equation was integrated by a velocity Verlet algorithm with a time step of δt=0.01τ, where $\tau =\sigma \sqrt{m/\epsilon }$. The particle mass *m* is arbitrary and does not affect the results. A constant temperature, T=300K, was set for all systems by applying the Langevin thermostat with damping constant $\gamma =1.0\tau^{-1}$. Duration of each simulation was $10^{5}$ cycles where each cycle consisted of 10 reaction moves followed by 100 integration steps of the Langevin dynamics. We disregard the first 20% of runs for the equilibration, and the remaining part was used for production and analysis. The total length of the simulation was adjusted to approximately $10^{3}$ uncorrelated samples of the radius of gyration. The correlation-corrected error estimates were used to assess the statistical accuracy of our data [62].

### 2.2. Atomistic Simulations

#### 2.2.1. Simulation Model

In our previous work we used all-atom simulations of amino acid tetramers shown in Figure 2a–d in order to determine average distances within the molecules which correspond to the distances $r_{C-C}$, $r_{A-C}$, $r_{B-C}$ (see Appendix A). We simulated only the fully ionised forms of these tetramers because interactions between the ionised groups play a key role in the charge regulation. To ensure the validity of our model, we verified that the average values of relevant charge-charge distances obtained from the all-atom simulations were reasonably well reproduced in the coarse-grained simulations.

In the current work we complemented the previous simulations with a simulation of His_4_ tetramer, shown in Figure 2b, solvated by 4028 water molecules in a cubic box length of L=6.00nm, which corresponds to the tetramer concentration of 31mM. The fully ionised tetramer was neutralized by adding ${\mathrm{Na}}^{+}$ ions. Additional salt ions in the system determined the ionic strength, corresponding to the salt concentration of 0.05M. In total $11{\mathrm{Cl}}^{-}$ and $7{\mathrm{Na}}^{+}$ ions were present in the simulation box. We used the Gromacs 2018.6 package to perform the all-atom simulations [63,64]. The average values of distances $r_{C-C}$ and $r_{A-C}$ obtained tom the simulations of His_4_, the corresponding probability distributions and comparison with the CG model are provided in the Appendix A.

#### 2.2.2. Interaction Potentials

AMBER99sb-ILDN force filed was used for the peptide tetramer, and TIP3P force field was used for the water molecules. All bonds were constrained with the LINCS algorithm and long-range electrostatic interactions were calculated using the Particle Mesh Ewald (PME) method. The Van der Waals interactions were truncated at 1.2 nm.

#### 2.2.3. Simulation Protocol and Data Analysis

We performed $5\times 10^{4}$ energy minimization steps, followed by 500ps[NVT] run, followed by 100ns[NPT] run. The last 90ns of the [NPT] run was used for production and data analysis. The steepest descent method was used for energy minimization. The velocity re-scaling algorithm was used for the [NVT] run, and the Parrinello-Rahman algorithm was used for the [NPT] run. We set the temperature to T=300K, and pressure to 1bar. The temperature coupling constant was τ=0.1ps, and the pressure coupling constant τ=2.0ps. The integration time step was 2fs for all simulations.

### 2.3. Experiments

The experiments described below follow the same protocol as in our previous publication [59]. Therefore, we only briefly outline the key features of these experiments, and refer the reader to the earlier publication for full technical details.

#### 2.3.1. Materials

Custom-synthesized peptides with acetyl and amide terminal groups and trifluoroacetate (TFA) as counterion were purchased from Biomatik LLC, Wilmington, Delaware, USA. All peptides were purified, and HPLC and MS spectra were measured for all peptide sequences by the producer. Standardised solutions of HCl and NaOH from Carl Roth GmbH (Karslruhe, Germany) were used to prepare 0.1M stock solutions for potentiometric titrations. These stock solutions were subsequently diluted to 0.01M. To prevent contamination by CO_2_, the standardised solutions were kept under soda lime at least 24 h before the measurements. Deuterium oxide 99.8% purity with a trace of 3-(trimethylsilyl)-1-propanesulfonic acid sodium salt (DSS) of 97% purity from Sigma-Aldrich was used for field-frequency lock in the NMR experiments.

#### 2.3.2. Potentiometric Titration

The peptides were dissolved in 0.01M standardised HCl to prepare solutions at final concentrations of monomeric units: [Tyr]=[Lys]=5mM, resulting in the concentration of peptide chains $c_{\mathrm{peptide}}=1\;\mathrm{mM}$. Sample volumes of approximately 2mL were weighed to determine the precise amount and then titrated with standardised 0.01M NaOH. Potentiometric titrations were performed using a Metrohm 888 Titrando Compact titrator equipped with a Metrohm LL Biotrode 3mm glass electrode, a Pt1000 temperature sensor, a titration vessel for 1mL, magnetic stirrer and Titrando Software.

The charge on the peptide was calculated from the measured pH and from the known volumes of HCl and NaOH
(9)$$z_{\mathrm{titration}}=\frac{V_{\mathrm{HCl}}c_{\mathrm{HCl}}-V_{\mathrm{NaOH}}c_{\mathrm{NaOH}}+\left(c_{\mathrm{OH}}-c_{H}\right)\left(V_{\mathrm{HCl}}+V_{\mathrm{NaOH}}\right)}{c_{\mathrm{peptide}}V_{\mathrm{HCl}}}+z_{\max}x_{\mathrm{TFA}},$$
where *c* is the concentration, *z* is the charge on the peptide, $z_{\max}=5$ and $x_{\mathrm{TFA}}≳1$ is the mole fraction of trifluoroacetate (TFA) counterions contained in the peptide sample, relative to the basic side chains on the peptide. The concentrations of H^+^ and OH^-^ ions, used in Equation (Equation 9), were calculated from the measured pH and $pK_{w}$. Because $pK_{w}$ is sensitive to temperature, $z_{\mathrm{titration}}\left(\mathrm{pH}\right)$ from Equation (Equation 9) was also sensitive to temperature and to the precision of the pH measurement, yielding reliable results only at intermediate pH, 3≳pH≳11. To correct for the unknown value of $x_{\mathrm{TFA}}$, we used it as an adjustable parameter to match the charge of the peptide at pH=6 (see the Appendix A for technical details). We quantified the reliability of $z_{\mathrm{titration}}\left(\mathrm{pH}\right)$ by comparing titrations of peptide samples with blank titrations of HCl stock solutions.

#### 2.3.3. NMR

All NMR data were recorded using a Bruker AVANCE III spectrometer operating at the proton Larmor frequency of 600MHz equipped with a cryogenically cooled probe and stabilising the temperature at $25\;{}^{\circ }C$. The samples for NMR were prepared by dissolving each peptide in 0.01M HCl to a final concentration of $15\;{\mathrm{gL}}^{-1}$ and by titrating the solutions with NaOH to adjust the pH to the desired value. 2D NMR spectra, NOESY and ${}^{1}$H-${}^{13}$C HSQC, at pH 2 were used for peak assignment (see Appendix A). The degrees of ionisation were determined from the chemical shift of specific atoms, which were identified in the literature as good reporters of ionisation [65]
(10)$$\alpha_{\mathrm{base}}\left(\mathrm{pH}\right)=\frac{\delta_{max}-\delta \left(\mathrm{pH}\right)}{\delta_{max}-\delta_{min}}\;,\;\alpha_{\mathrm{acid}}\left(\mathrm{pH}\right)=\frac{\delta \left(\mathrm{pH}\right)-\delta_{min}}{\delta_{max}-\delta_{min}}.$$

For all 5 amino acid signals we calculated a single value of the chemical shift, determined by the centre of mass of the corresponding peak. Detailed information about ionisation of each individual group might be possibly extracted from these peaks [66,67]. However, we did not attempt to resolve such details because it would go beyond the scope of the current study. The MestReNova Software was used to analyse both 1D and 2D spectra, including the determination of centers of mass of multiplets and the ranges of the peaks.

## 3. Results and Discussion

### 3.1. Charge Regulation

In Figure 3, Figure 4 and Figure 5 we show the total charge on various peptides as a function of pH. Each of these peptides consists of five amino acids with acidic side chains and five with basic side chains but they differ in the $pK_{A}^{\mathrm{acid}}$ and $pK_{A}^{\mathrm{base}}$ of the side chains. The values of $pK_{A}^{\mathrm{acid}}$ and $pK_{A}^{\mathrm{base}}$ were chosen by combining pairs of amino acids listed in Table 1, such that they include the following values of $\Delta pK_{A}\equiv pK_{A}^{\mathrm{base}}-pK_{A}^{\mathrm{acid}}$: (1) $\Delta pK_{A}\ll 0$, represented by Tyr + His in Figure 3; (2) $\Delta pK_{A}\gg 0$, represented by Lys + Asp in Figure 4; (3) $\Delta pK_{A}\approx 0$, represented by Glu + His and Tyr + Lys in Figure 5 and $\Delta pK_{A}=0$, represented by hypothetical Acid + Base in Figure 5. For each combination of $pK_{A}^{\mathrm{acid}}$ and $pK_{A}^{\mathrm{base}}$, we compared two kinds of sequences: diblock or alternating, as shown in Figure 1 and Table 2. Each of them was compared to the ideal ionisation calculated from the Henderson-Hasselbalch equation, Equation (Equation 1). As seen in Figure 3a, Figure 4a and Figure 5a,c,e, the diblock and alternating sequences exhibit very similar deviations from the ideal ionisation response. We will show later that this similarity is a result of non-trivial cancellation of rather strong non-ideal effects. To understand these effects, we first discuss the role of $\Delta pK_{A}$, and then we discuss the role of sequence for each value of $\Delta pK_{A}$.

#### 3.1.1. $\Delta pK_{A}\ll 0$

In the case of $\Delta pK_{A}\ll 0$, (Figure 3a) both acid and base groups are uncharged at the isoelectric point. When moving from the isoelectric point towards higher pH values, the base groups remain uncharged, while the acid groups acquire negative charge (Figure 3b), resulting in an overall negative charge on the whole peptide (Figure 3a). The repulsion between these charges of equal signs provides a barrier which must be overcome when increasing the charge on the peptide chain. This effect is well known from previous studies of synthetic polyelectrolytes consisting of charges of one type [8]. When moving from the isoelectric point towards lower pH values, the acid groups remain uncharged, while the base groups acquire positive charge, resulting in the same effect as in the case of high pH, except that now it is caused by the positively charged base groups. Thus, in both cases the magnitude of the charge on the peptide is lower than that predicted by the Henderson-Hasselbalch equation (Figure 3a).

When comparing the ionisation of diblock and alternating peptides in Figure 3b, we observe that deviation from the HH equation is greater in the case of diblock. This can be understood by realizing that ionisable groups are closer to each other in the diblock, resulting in a stronger electrostatic repulsion. In the alternating peptide, the neighbouring acid groups are separated by the base groups, and the latter are neutral in the pH range where the acid is ionised. Again, this observation is consistent with previous studies of weak polyelectrolytes, where it has been observed that the deviations from the ideal ionisation decrease with increasing separation of the ionisable groups.

As a general rule, explanations for the observed effects at pH>pI and pH<pI are the same, except that the roles of acid and base groups are swapped. Therefore, in the remaining discussion we will describe only the case of pH>pI, assuming that explanation for the other case is implied. However, we note that our results are not completely symmetric around pH=pI because of slightly different parameters used to describe individual amino acid side chains.

#### 3.1.2. $\Delta pK_{A}\gg 0$

The total charge on the peptide in the case of $\Delta pK_{A}\gg 0$ (Figure 4a) is a very similar picture to the case of $\Delta pK_{A}\ll 0$, (Figure 3a). However, in the former case both types of ionisable groups (acid and base) are fully charged at the isoelectric point (Figure 3a), whereas in the latter case both types are uncharged at the isoelectric point. Because the absolute value of $\Delta pK_{A}$ of ${\mathrm{Lys}}_{5}-{\mathrm{Asp}}_{5}$ is much greater than that of ${\mathrm{Tyr}}_{5}-{\mathrm{His}}_{5}$, the ${\mathrm{Lys}}_{5}-{\mathrm{Asp}}_{5}$ peptide is neutral in a rather broad range of pH values. When moving from the isoelectric point towards higher pH values, the acid groups remain fully charged (Figure 4b), while the base groups lose their charge as the pH increases. The net result is that the peptide acquires an overall negative charge, similar to what we observed in Figure 3a for the ${\mathrm{Tyr}}_{5}-{\mathrm{His}}_{5}$ peptide.

Even though the total charge on the peptide has a lower magnitude than predicted by the HH equation in both cases, $\Delta pK_{A}\gg 0$ and $\Delta pK_{A}\ll 0$, the physical reasons for this observation are different. At $\Delta pK_{A}\gg 0$ the observation cannot be explained by simple analogy with weak polyelectrolyte homopolymers, as it was in the case of $\Delta pK_{A}\ll 0$. Figure 4b reveals that at $\Delta pK_{A}\gg 0$ the charge of the ionisable groups is not lower but higher than that predicted by the HH equation. Thus, at high pH values, the base groups lose less charge than one would expect based on the Henderson-Hasselbalch equation. The net charge at $\Delta pK_{A}\gg 0$ is given by the difference between the fully charged acid groups and partly charged base groups. Therefore, the enhanced ionisation of acid and base groups at $\Delta pK_{A}\gg 0$ results in the same effect on the net charge of the peptide, as their suppressed ionisation at $\Delta pK_{A}\ll 0$.

When comparing the ionisation of diblock and alternating peptides in Figure 4b, we observe that ionisation of the base groups on the alternating peptides is enhanced more than ionisation of the same groups on the diblock. This difference can be explained by oppositely charged acid groups located in the immediate vicinity of the base groups in the alternating peptide sequence. The diblock peptide contains the same amount of oppositely charged acid groups, however, they are located further from the base groups, resulting in a weaker effect.

#### 3.1.3. $\Delta pK_{A}\approx 0$ and $\Delta pK_{A}=0$

The case $\Delta pK_{A}\approx 0$ and $\Delta pK_{A}=0$ is presumably the most interesting because the ionisation of acid and base groups changes simultaneously in the same range of pH, and these groups mutually influence each other’s ionisation. In addition to peptides based on real amino acids with $\Delta pK_{A}=1.75≳0$ (Glu + His) and $\Delta pK_{A}=0.47≲0$ (Tyr + Lys), we also investigated peptides with $\Delta pK_{A}=0$ {${\mathrm{Acid}}_{5}-{\mathrm{Base}}_{5}$, ${\left(\mathrm{Acid}-\mathrm{Base}\right)}_{5}$}, composed of hypothetical amino acids (a + b). Results for each of these systems are similar but the case of $\Delta pK_{A}=0$ more clearly demonstrates the cancellation of effects that we observe here. Therefore, unless explicitly specified, we will not distinguish between these three cases in the discussion, referring to all of them as $\Delta pK_{A}\approx 0$.

The total charge on the peptide in the case of $\Delta pK_{A}\approx 0$ (Figure 5a,c,e) is lower than the charge predicted by the HH equation, same as in the previously discussed cases, $\Delta pK_{A}\gg 0$ and $\Delta pK_{A}\ll 0$. However, in this case the situation is most complicated, combining the antagonistic effects observed in the two extreme cases of $\Delta pK_{A}$. As $\Delta pK_{A}$ approaches zero, the pH region in which the peptide is neutral shrinks to zero as well, and at $\Delta pK_{A}=0$ the charge as a function of pH is actually varying most steeply at pH=pI. This is in contrast with the previously discussed cases, $\Delta pK_{A}\gg 0$ and $\Delta pK_{A}\ll 0$, where the charge as a function of pH was almost constant in a rather broad region of pH≈pI.

In the case of $\Delta pK_{A}\approx 0$, both acid and base groups are only partly ionised at the isoelectric point and both exhibit opposite deviations from the ideal behaviour. Also, in this case the ionisation response of acid and base groups in the diblock and alternating sequence becomes qualitatively different. Ionization response of individual groups in the diblock sequence is symmetric around pH=pI. At pH>pI, ionisation of the acid groups in the diblock sequence is suppressed and simultaneously ionisation of the base groups is enhanced as compared to the HH equation. Their combined effect on the net charge results in deviations from the HH equation which are very similar to the cases of $\Delta pK_{A}\gg 0$ and $\Delta pK_{A}\ll 0$. In contrast with that, ionisation response of the alternating sequence is not completely symmetric around pH−pI. At pH>pI, ionisation of the acid groups in the alternating sequence is enhanced even at pH≳pI, which was not observed in any of the previously discussed cases. This situation changes at pH>pI+1, when ionisation of the acid is eventually suppressed as compared to the HH equation. In the same range of pH, ionisation of the base groups is enhanced. Analogous arguments hold for the range pH<pI, except that the roles of acid and base groups are swapped.

To conclude the discussion of charge regulation, we observed that the value of $\Delta pK_{A}$ seems to have very little effect on the magnitude of the deviations of total charge on the peptide from the ideal behaviour. Nevertheless, the physical reasons for these deviations are different in each case, resulting in more subtle differences in the ionisation of individual groups. When comparing the charge on the diblock and alternating peptides, $\Delta pK_{A}>0$ results in the charge on the diblock being closer to the ideal charge than alternating sequence. At $\Delta pK_{A}>0$ this situation is reversed, resulting in charge on the alternating sequence deviating from the ideal behaviour more than the diblock. In the hypothetical case of $\Delta pK_{A}=0$, these effects quantitatively cancel, resulting in the net charge on both sequences being identical within the statistical error of our simulations.

### 3.2. Experimental Validation

To check to what extent our simulation predictions are reliable, we validated them against experimentally determined charge and ionisation degree of real peptides. For ${\mathrm{Glu}}_{5}-{\mathrm{His}}_{5}$ and ${\mathrm{Lys}}_{5}-{\mathrm{Asp}}_{5}$, such experimental validation has been published in our previous study [59]. In that study, the simulated ionisation degree of individual amino acid side-chains was well matched by the ionisation degree determined from NMR chemical shifts. In addition, the total charge on both peptides was well matched not only by NMR data but also by potentiometric titration and capillary zone electrophoresis (CZE). From these results we also learned that even though the shape of the curves well agrees between the simulations and experiments, they can be mutually shifted because $pK_{A}$ of the side chain is affected by the change of local substituents upon incorporation of the amino acid in the peptide. Because this change of $pK_{A}$ cannot be predicted by coarse-grained simulations, we used $pK_{A}$ of free amino acids as simulation inputs. This resulted in systematic shifts on the pH scale between the simulations and experiments, and these shifts decreased as the distance of the ionisable group from the peptide backbone increased. The biggest shift was observed for Asp, smaller shifts were observed for Glu and His, while almost no shift was observed for Lys.

In the current study, we supplement the earlier experimental results by the experiments on ${\mathrm{Tyr}}_{5}-{\mathrm{Lys}}_{5}$, shown in Figure 6. CZE experiments could not be performed for this peptide because of its tendency to stick to the capillary walls. Therefore, we compared the simulations only with NMR results and potentiometric titration. In addition, the absence of CZE results did not allow us to correct for the excess of TFA counterions which were causing a systematic shift in the potentiometric titration curves. Therefore, we corrected for this excess of TFA by shifting the titration curves to yield the maximum peptide charge, z=+5 at pH=5, as detailed in the Appendix A. The comparison of total charge in Figure 6a reveals a quantitative agreement between the simulations and titrations, whereas the agreement with NMR seems worse. Nevertheless, considering the rather big uncertainty in the charge determined from NMR, we may still conclude that simulation predictions are in agreement also with this experimental method. The uncertainty in the NMR results is caused by the simplified method we used to extract a single chemical shift value representing the state of all 5 amino acids of a given type. In fact, each amino acid provides one peak in the 13C NMR spectrum and these peaks shift and overlap and broaden as pH changes the ionisation state of each particular amino acid. We evaluated the center of gravity of the group of corresponding 5 peaks and neglected difference in the behavior of individual peaks, presumably caused by different local effects on the ionisation. Furthermore, comparison of the ionisation degree of acid and base groups with simulation results reveals the same trend as described in the previous study [59]: overall shape of the curves is reproduced but the experimental values are shifted to higher pH, which is presumably caused by the effect of local substituents. Thus, we conclude that the simulation predictions of both total charge and ionisation degree on acid and base groups well match the values determined experimentally. Some mismatch in the ionisation of Lys could be attributed to experimental uncertainty, whereas mismatch in the ionisation of Tyr could be explained by the shift of $pK_{A}$ upon incorporation of amino acid in the peptide.

### 3.3. Chain Conformations

In Figure 7, we show the average distances between the first and last central beads (end-to-end distance, $R_{e}$) of various peptides as a function of pH. All end-to-end distances attain a maximum at extreme pH values because one type of the ionisable groups (acid or base) is fully ionised while the other type is neutral. In the diblock sequences, this ionisation state results in a tadpole conformation, where the charged block is stretched while the neutral block is coiled. This is supported by simulation snapshots in Figure 8, and it is also consistent with observations in other studies [44,46,55]. In the alternating sequences, this ionisation state results in a uniformly stretched conformation, where like-charged groups of one type are separated by neutral groups of the other type. This uniform stretching causes that the absolute value of $R_{e}$ at extreme pH values is always greater in the alternating sequence than in the diblock sequence.

The end-to-end distances attain a minimum around the isoelectric point for each peptide, irrespective of its $\Delta pK_{A}$ or amino acid sequence (diblock or alternating). However, in the diblock sequences the value of $\Delta pK_{A}$ determines how much $R_{e}$ at the isoelectric point differs from $R_{e}$ at high or low pH. If $\Delta pK_{A}≳0$, then both acid and base groups are ionised at the isoelectric point. Their mutual attraction results in a more compact conformation than in the case when only groups of one type were ionised. On the other hand, if $\Delta pK_{A}\ll 0$, then all groups are neutral at the isoelectric point, and their electrostatic interaction does not significantly affect the conformation. Consequently, as a general trend, the minimum of $R_{e}\left(\mathrm{pH}\right)$ at the isoelectric point is well pronounced at $\Delta pK_{A}>0$ and it diminishes as $\Delta pK_{A}$ is decreased towards $\Delta pK_{A}<0$. This general trend is much less significant in the alternating sequences because the alternating positive and negative charges effectively screen each other. Consequently, their attractions and repulsions roughly cancel at the isoelectric point, irrespective of $\Delta pK_{A}$.

Furthermore, comparison of Figure 7a,b reveals that the minimum of $R_{e}$ at the isoelectric point is much more pronounced for the diblock sequences than for the alternating ones. This is because the diblock sequence can benefit from favourable electrostatic interactions only if the oppositely charged acid and base blocks are brought close to each other as a consequence of a conformational change. However, the absolute value of $R_{e}≳1.4\;\mathrm{nm}$ suggests that the oppositely charged blocks do not completely fold on top of each other. On the contrary, opposite charges in the alternating sequence are next to each other, therefore a change in conformation does not bring such a strong gain in the electrostatic interaction energy as in the diblock sequence. Thus, we have shown that even though diblock and alternating sequences of the same amino acids have almost identical ionisation response to pH, their conformational response differs significantly. We anticipate that in both cases the conformational changes become more pronounced for longer peptide sequences, which will be discussed in the next section.

### 3.4. The Role of Chain Length

To investigate the role of chain length, we have chosen only the ${\mathrm{Lys}}_{n}-{\mathrm{Asp}}_{n}$ and ${\left(\mathrm{Lys}-\mathrm{Asp}\right)}_{n}$ peptide sequences because the ionisation of their acid and base blocks varies in different ranges of pH, which allows us to demonstrate the analogy with synthetic polyelectrolytes. For synthetic weak polyelectrolytes it is known that the shift in their ionisation response increases as the chain length increases, especially for rather short chains, consisting of N≲50 ionisable groups [31]. Our model peptides exhibit a similar behaviour as synthetic polyelectrolytes, shown in Figure 9. Deviation of the ionisation of acid and base groups on each peptide from the ideal curve increases with increasing chain length. Simultaneously, we observe again that there is very little or no difference between the ionisation response of diblock and alternating sequences.

In contrast with the ionisation response, the differences between the conformations of diblock and alternating sequences become more pronounced for longer chains. In Figure 10 we observe that all diblock sequences have a very small value of $R_{e}<3\;\mathrm{nm}$ at the isoelectric point. The simulation snapshots in Figure 8 reveal that this is because they form a compact globular conformation, resembling droplets of complex coacervates. This conformation allows the diblock sequence to fully exploit the enthalpy gain due to electrostatic attraction of oppositely charged groups. In contrast with that, $R_{e}$ values of the alternating sequences are much higher, and they also increase more significantly as the chain length increases. This is consistent with our previous claim that the alternating sequence does not need a big conformational change to benefit from the electrostatic attraction. Therefore, the alternating sequence has a much lower tendency to form a droplet-like compact structure even if the chain is rather long. This observation well compares with related studies of the stability of coacervates formed by ampholytes with various sequences of charges [51,54,55].

## 4. Conclusions

In this study we systematically investigated the role of amino acid sequence in the charge regulation and concomitant conformational changes of various oligopeptides in a broad range of pH. Each of the model peptides consisted of two kinds of amino acids: one with acidic side chains, the other one with basic side chains. Different combinations of amino acids allowed us to systematically vary the difference in their $pK_{A}$ values, $\Delta pK_{A}=pK_{A}^{\mathrm{base}}-pK_{A}^{\mathrm{acid}}$, covering the whole range from $\Delta pK_{A}\gg 0$, through $\Delta pK_{A}\approx 0$ to $\Delta pK_{A}\ll 0$.

In peptides with $\Delta pK_{A}\gg 0$ both acid and base groups are fully ionised at the isoelectric point and one type of the groups loses its charge when moving from the isoelectric point towards extreme pH values. On the contrary, in peptides with $\Delta pK_{A}\ll 0$, both types of groups are neutral at the isoelectric point, and one type of the groups gains charge when moving towards extreme pH values. Finally, in peptides with $\Delta pK_{A}\approx 0$ the changes of ionisation of acid and base groups proceed simultaneously in the pH range close to the isoelectric point. Electrostatic interactions between the anionic acid groups and cationic base groups cause significant deviations from the ideal ionisation response predicted by the Henderson-Hasselbalch equation. Even though the simulated curves retain the qualitative features of the ideal curves, the real ionisation response is shifted away from the isoelectric point, towards the extreme pH values. In the case of $\Delta pK_{A}\gg 0$, this shift is caused by electrostatic attraction with the oppositely charged groups, resulting in an enhanced ionisation. In the case of $\Delta pK_{A}\ll 0$ this shift is caused by electrostatic repulsion between the like-charged groups of the same type, resulting in a suppressed ionisation. Interestingly, these shifts in the ionisation response are caused by opposite effects but they also affect charges of opposite signs. Therefore, they both lead to a decreased total charge on the peptide, as compared to the charge expected from the ideal ionisation.

Furthermore, for each value of $\Delta pK_{A}$ we compared the behaviour of diblock and alternating sequences. Our simulations showed that the ionisation response of diblock and alternating sequences is almost indistinguishable, caused by a similar cancellation of effects as mentioned above. However, despite the same total charge, the alternating and diblock sequences exhibit quite different conformational response to a change in pH. At extreme pH values the diblocks form tadpole conformations with one block stretched and the other one coiled, while the alternating ones form uniformly stretched chains. Close to the isoelectric point, the diblocks form compact structures, resembling droplets of complex coacervates in order to minimize the distance between the oppositely charged groups. On the contrary, alternating sequences are much more loose at the isoelectric point because the proximity of opposite charges is already ensured by the sequence and favourable electrostatic interactions do not require such dramatic conformational changes.

Finally, we looked into the role of chain length, using just one selected peptide. We observed that as the chain length is increased, the magnitude of all effects increases, while simultaneously retaining all their qualitative features. The short peptide sequences used in the first part of the study have been chosen with the intention to facilitate direct comparison with experiments. Such comparison would be much more complicated for longer chains because of higher uncertainty of the experimentally determined charge or ionisation degree. For three of the simulated peptide sequences our parameter-free comparison with experimental results revealed excellent agreement of the simulation predictions with experiments (two of these have been reported in previous study [59]). This agreement provides a very promising perspective of using similar modeling techniques to predict charge regulation and concomitant conformational changes not only in short oligopeptides but also in flexible disordered proteins or synthetic ampholytes.

## Figures and Tables

**Figure 1 polymers-13-00214-f001:**
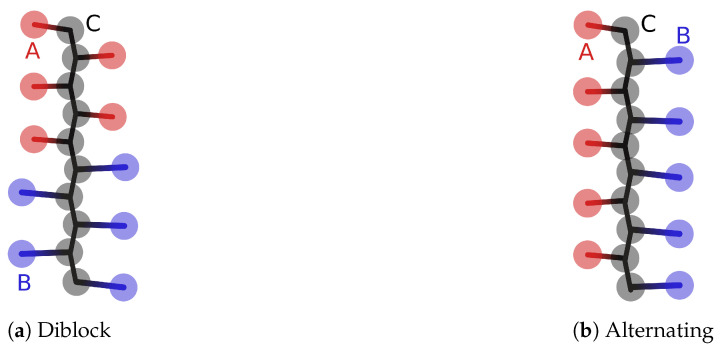
Schematic representation of the coarse-grained bead-spring model for the diblock peptides (**a**) and alternating peptides (**b**). The A beads represent acid side chains, B beads represent base side chains, and central beads C represent the amino acid backbone composed of the peptide bonds.

**Figure 2 polymers-13-00214-f002:**
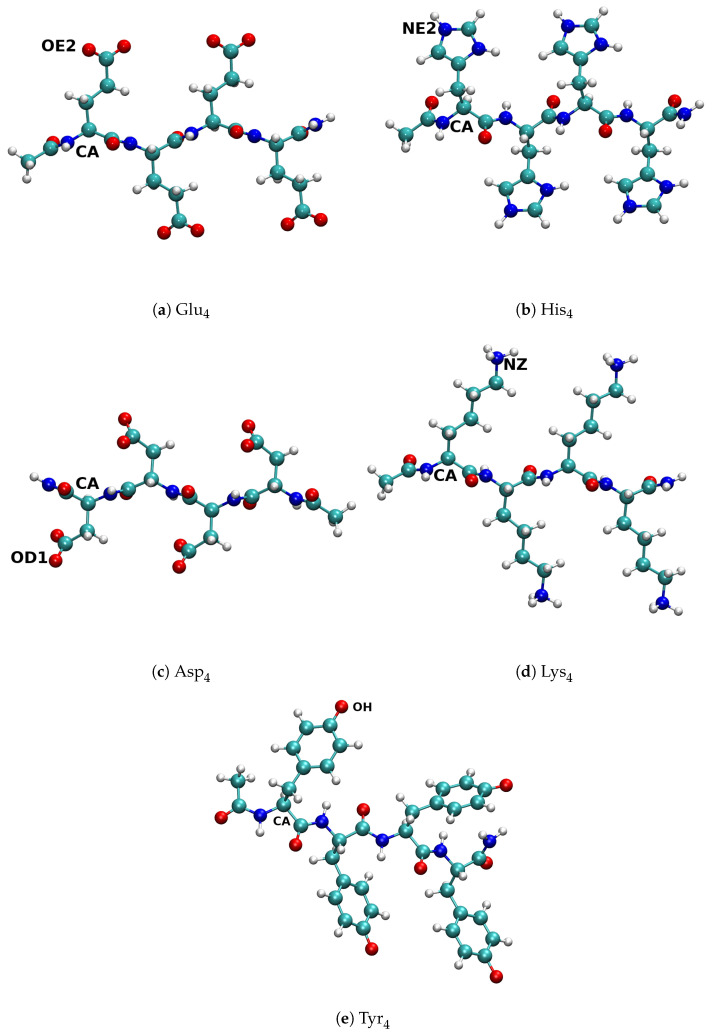
Initial configurations of the peptides used in all-atom (AA) simulations. Labels mark the atoms which we used to measure various intra-molecular distances.

**Figure 3 polymers-13-00214-f003:**
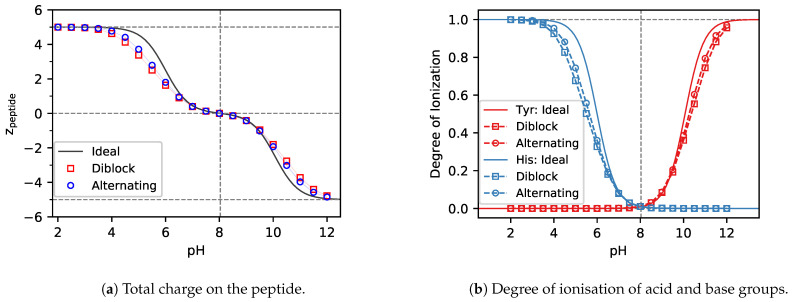
Simulation predictions of the total charge and ionisation degree of the ${\mathrm{Tyr}}_{5}-{\mathrm{His}}_{5}$ and ${\left(\mathrm{Tyr}-\mathrm{His}\right)}_{5}$ peptides with $\Delta pK_{A}=-4.07$. Solid lines represent the ideal result from the Henderson-Hasselbalch equation. Squares represent the diblock and circles the alternating sequence.

**Figure 4 polymers-13-00214-f004:**
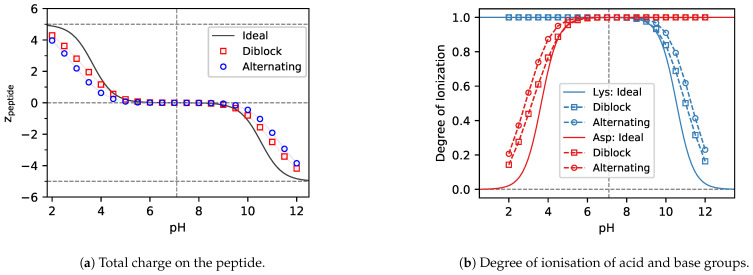
Simulation predictions of the total charge and ionisation degree of the ${\mathrm{Lys}}_{5}-{\mathrm{Asp}}_{5}$ and ${\left(\mathrm{Lys}-\mathrm{Asp}\right)}_{5}$ peptides with $\Delta pK_{A}=6.89$. Solid lines represent the ideal result from the Henderson-Hasselbalch equation. Squares represent the diblock and circles the alternating sequence.

**Figure 5 polymers-13-00214-f005:**
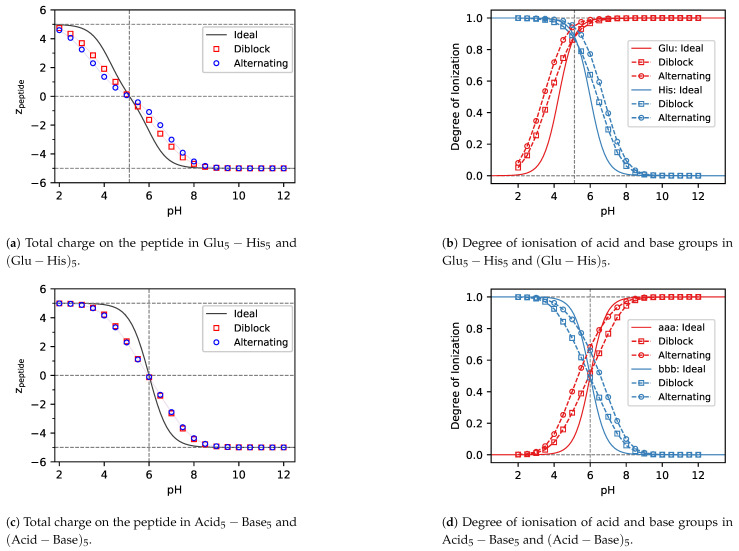
Simulation predictions of the total charge and ionisation degree of the ${\mathrm{Glu}}_{5}-{\mathrm{His}}_{5}$ and ${\left(\mathrm{Glu}-\mathrm{His}\right)}_{5}$ peptides with $\Delta pK_{A}=1.75$, the ${\mathrm{Tyr}}_{5}-{\mathrm{Lys}}_{5}$ and ${\left(\mathrm{Tyr}-\mathrm{Lys}\right)}_{5}$ peptides with $\Delta pK_{A}=0.47$, and hypothetical peptides ${\mathrm{Acid}}_{5}-{\mathrm{Base}}_{5}$ and ${\left(\mathrm{Acid}-\mathrm{Base}\right)}_{5}$ with $\Delta pK_{A}=0$. Solid lines represent the ideal result from the Henderson-Hasselbalch equation. Squares represent the diblock and circles the alternating sequence.

**Figure 6 polymers-13-00214-f006:**
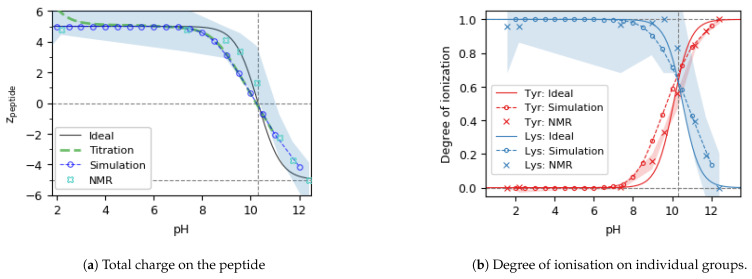
Total charge (**left**) and degree of ionisation (**right**) of the peptides as a function of pH from the ideal Henderson-Hasselbalch equation, simulations, and NMR. The gray and red vertical lines indicates the isoelectric point.

**Figure 7 polymers-13-00214-f007:**
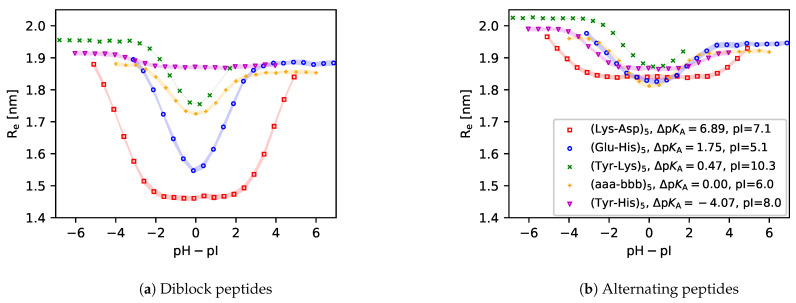
Simulation predictions of the end to end distance of various peptide sequences.

**Figure 8 polymers-13-00214-f008:**
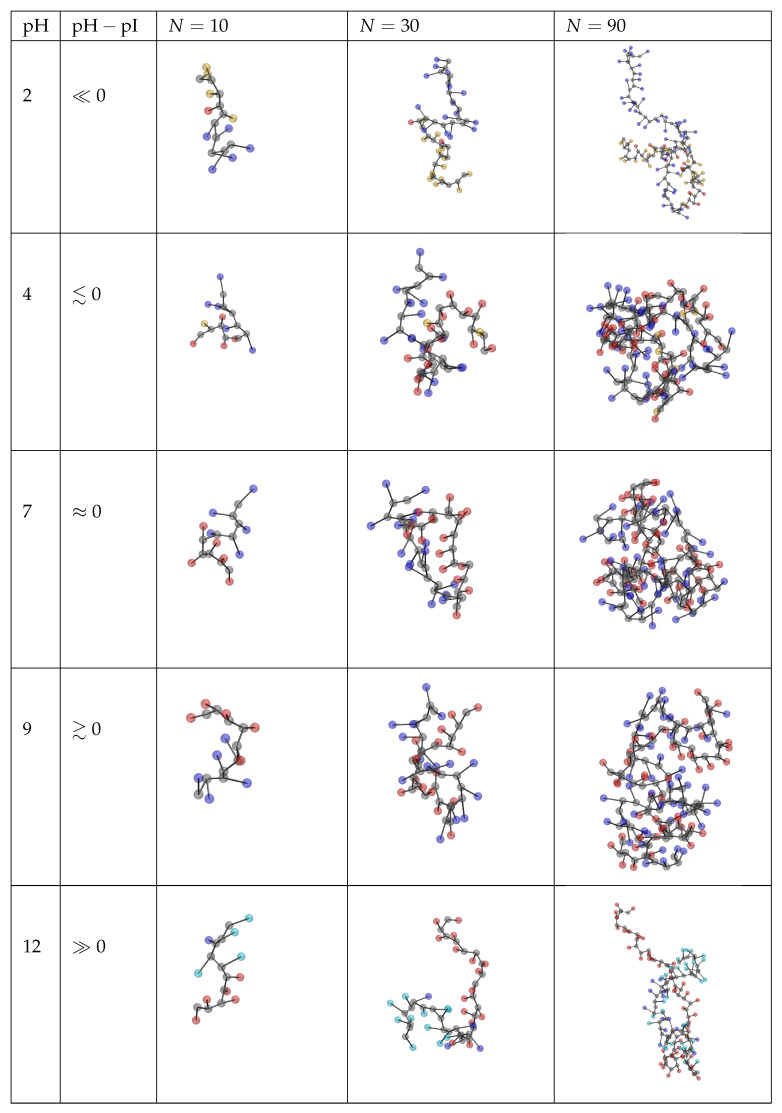
Simulation snapshots of the ${\mathrm{Lys}}_{5}-{\mathrm{Asp}}_{5}$ peptide at selected values of pH. Colour code: grey = backbone, red = ionised acid group, yellow = non-ionised acid group, blue = ionised base group, cyan = non-ionised base group.

**Figure 9 polymers-13-00214-f009:**
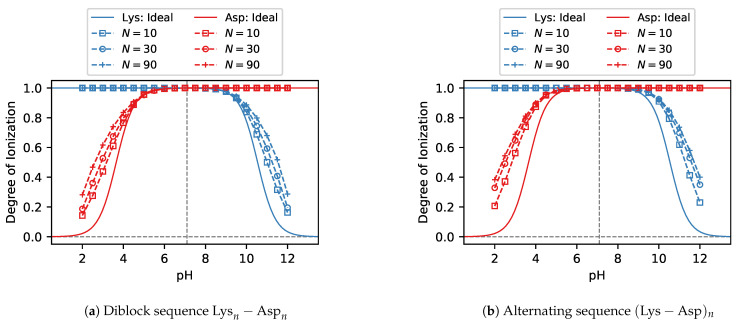
Simulation predictions of the ionisation degree in (**a**) Diblock peptides ${\mathrm{Lys}}_{n}-{\mathrm{Asp}}_{n}$ and (**b**) Alternating peptide peptides ${\left(\mathrm{Lys}-\mathrm{Asp}\right)}_{n}$ with various lengths of the sequences N=2n.

**Figure 10 polymers-13-00214-f010:**
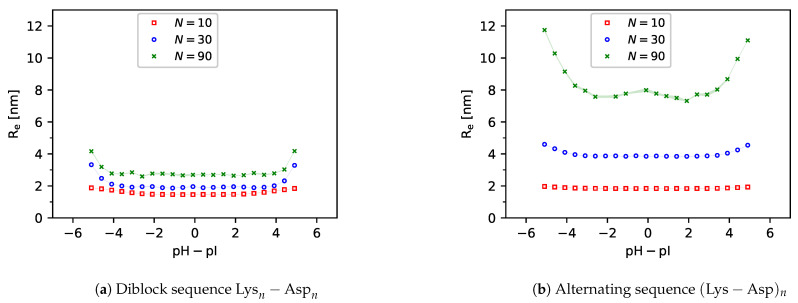
Simulation predictions of the end-to-end distances in (**a**) Diblock peptides ${\mathrm{Lys}}_{n}-{\mathrm{Asp}}_{n}$ and (**b**) Alternating peptide peptides ${\left(\mathrm{Lys}-\mathrm{Asp}\right)}_{n}$ with various lengths of the sequences N=2n.

**Table 1 polymers-13-00214-t001:** Parameters of the amino acids used in the simulations: $pK_{A}$ value of the ionisable side chain [60] and the average distance between the central and the side-chain beads (AC or BC, see also Figure 1). a and b denote hypothetical amino acids with acid and base side-chains, which have the same $pK_{A}$ values.

	One-Letter		Bond Length	Ionizable
Full Name	Code	$pK_{A}$	AC or BC [nm]	Group Type
Aspartic acid	D	3.65	0.327±0.029	carboxyl
Glutamic acid	E	4.25	0.436±0.044	carboxyl
Histidine	H	6.00	0.453±0.013	1H-imidazol-4-yl
Tyrosine	Y	10.07	0.648±0.012	4-hydroxyphenyl
Lysine	K	10.54	0.589±0.042	amine
hypothetical Acid	a	6.00	0.355±0.000	–
hypothetical Base	b	6.00	0.355±0.000	–

**Table 2 polymers-13-00214-t002:** Parameters of various peptides simulated in this study: $\Delta pK_{A}$ values and amino acid sequences. For parameters of individual amino acids see Table 1.

		Diblock		Alternating	
$\Delta pK_{A}$	Ideal pI	Abbreviated Name	Sequence	Abbreviated Name	Sequence
6.89	7.1	${\mathrm{Lys}}_{5}-{\mathrm{Asp}}_{5}$	KKKKKDDDDD	${\left(\mathrm{Lys}-\mathrm{Asp}\right)}_{5}$	KDKDKDKDKD
1.75	5.1	${\mathrm{Glu}}_{5}-{\mathrm{His}}_{5}$	EEEEEHHHHH	${\left(\mathrm{Glu}-\mathrm{His}\right)}_{5}$	EHEHEHEHEH
0.47	10.3	${\mathrm{Tyr}}_{5}-{\mathrm{Lys}}_{5}$	YYYYYKKKKK	${\left(\mathrm{Tyr}-\mathrm{Lys}\right)}_{5}$	YKYKYKYKYK
0	6	${\mathrm{Acid}}_{5}-{\mathrm{Base}}_{5}$	aaaaabbbbb	${\left(\mathrm{Acid}-\mathrm{Base}\right)}_{5}$	ababababab
−4.07	8	${\mathrm{Tyr}}_{5}-{\mathrm{His}}_{5}$	YYYYYHHHHH	${\left(\mathrm{Tyr}-\mathrm{His}\right)}_{5}$	YHYHYHYHYH

## Data Availability

Relevant data is included in the article. Raw data from measurements and simulations can be made available upon request from the corresponding author.

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
