# Peer review of "Role of pKA in Charge Regulation and Conformation of Various Peptide Sequences"

_polymers, 2021, doi:10.3390/polym13020214_

Round 1

Reviewer 1 Report

Dear editor:

I have read carefully the manuscript “Role of pKA in charge regulation and conformation of various peptide sequences” and I think that should be published with minor revisions. It represents a nice and significant contribution to the understanding of charge regulation in polyampholyte proteins. Moreover, the document is clear and well written and in my opinion it could be interest of “Polymer” readers.  

The authors investigate, by means of coarse-graning simulations, the influence of the ionization and conformational properties of two crucial trends of polyampholytes:

  1. i) the role of the distribution of acidic and basic groups along the chain. In this respect, they consider two limit situations: diblock location (acid-acid-…-acid-base-base-base-…-base) and the alternating location (acid-base-acid-base-…)
  2. ii) the difference between the pK-values of the acidic and basic groups (pKbase-pKacid)

They also compare their simulation results with experimental data, which makes the document even more interesting.

I have some remarks, questions and suggestions that I think should be clarified/discussed:

1) In the introduction, second paragraph, the sentence “Lastly, the term nanobuffering has been coined recently to describe systems in which the charge regulation is coupled to a conformational change in such a way, that it keeps the local environment approximately constant even though the pH-solution is changing”

I find the expression “it keeps the local environment approximately constant” ambiguous. What is exactly the authors try to explain? No macromolecular charge is produced in a part of the macromolecule? Or is the solution surrounding the macromolecule which has a stable pH-value?

2) In the coarse-graining model, the ionisable groups are located in lateral positions. Is that essential for the conclusions of the paper? If the groups had been located (to simplify more the conformational model) on the macromolecular backbone, the conclusions would have been different?

3) At the beginning when reading the document, I got little confused in the understanding of the work because the pKa-value of a basic group (Histidine, pKa=6) was smaller than the pKa-value of an acidic group (Tyrosine, containing a phenol group with pKa=10.07). This is not very usual in synthetic polymers, so I had to find by myself the composition of the different aminacids. I think it would help to follow the text (for the readers not familiar with polypeptides) to add a column in table 2 explicitly indicating the precise chemical group (amine, carboxylic, phenolic, etc) involved.

4) In order to find the distances involved in the coarse-graining model, AA simulations are performed. In particular, the fully ionized tetramer His4 is simulated. Why simulating the fully charge molecule and not the neutral one, or even intermediate ionization states? Is there some particular reason concerning the objectives of the work?

5) In analysing the NMR spectra, the expression (10) is used, which contains a fundamental approximation: the shift of a peak depends on the ionization state of the nearest group. It has been shown, however, that, in general, this is not the case, and the chemical shift is strongly coupled to more distant sites via electrostatic interaction with the nearest site (see, for instance, Madurga et al. J. Phys. Chem. A 2017, 121, 31, 5894–5906). As a consequence, the shift associated to a group depends on the ionization degree of the rest of the groups (or at least the closest ones). A general and rigorous treatment of the problem, which provides huge information about the site-specific ionization properties, can be found in Borkovec and Koper, Anal. Chem. 2000, 72, 14, 3272–3279. My question is: is the approximation involved in (10) suitable for poly-peptides? Is there some experimental evidence?  I think the use of equation (10) as an approximation should be at least mentioned.

6) Finally, two comments about writing:

In the second paragraph of section 3.3, it is written “the isoelectric point is well pronounced at ΔpKA>0, and it diminishes as ΔpKA is increased towards ΔpKA>0”. Is this sentence correct? It sounds contradictory.

In the captions of figure 1, the verb “represent” should be “represents”.

Reviewer 2 Report

The manuscript "Role of pKA in charge regulation and conformation of various peptide sequences" by Lunkad and co-authors is devoted to the computational study of model peptides and the influence of a type and position of charged groups in weak ampholytes on the overall molecule charge and conformation. The article is written in a consistent way and provide a methodic for the estimation of relative relationship in the systems with a higher complexity as disordered proteins.  

I have just a couple of minor comments:

  1. In the equation (I) the parameters are not well explained.
  2. I could not find a description of "hypothetical" ampholyte made of a and b, the partial charges, the distance between backbone and side chains beads, etc. Also it remains unclear what was a significance of its use.
  3. Please, check the subdivision In Methods. Besides, it is indicated that 2D NMR spectra (NOESY and HSQC) are used for the peak assignment. Did the authors try to extract an information about spatial organization of the peptides by using this techniques?
